# Efficacy and Safety of a Modified Vaccinia Ankara-NP+M1 Vaccine Combined with QIV in People Aged 65 and Older: A Randomised Controlled Clinical Trial (INVICTUS)

**DOI:** 10.3390/vaccines9080851

**Published:** 2021-08-03

**Authors:** Chris Butler, Chris Ellis, Pedro M. Folegatti, Hannah Swayze, Julie Allen, Louise Bussey, Duncan Bellamy, Alison Lawrie, Elizabeth Eagling-Vose, Ly-Mee Yu, Milensu Shanyinde, Catherine Mair, Amy Flaxman, Katie Ewer, Sarah Gilbert, Thomas G. Evans

**Affiliations:** 1Nuffield Department of Primary Health Care Sciences, University of Oxford, Oxford OX2 6GG, UK; christopher.butler@phc.ox.ac.uk (C.B.); hannah.swayze@phc.ox.ac.uk (H.S.); julie.allen@phc.ox.ac.uk (J.A.); ly-mee.yu@phc.ox.ac.uk (L.-M.Y.); milensu.shanyinde@phc.ox.ac.uk (M.S.); 2Vaccitech Ltd., Oxford OX4 4GE, UK; chris.ellis@vaccitech.co.uk (C.E.); louise.bussey@vaccitech.co.uk (L.B.); elizabeth.eagling-vose@vaccitech.co.uk (E.E.-V.); 3Jenner Institute, University of Oxford, Oxford OX3 7DQ, UK; pedro.folegatti@ndm.ox.ac.uk (P.M.F.); duncan.bellamy@ndm.ox.ac.uk (D.B.); alison.lawrie@ndm.ox.ac.uk (A.L.); 2177110M@student.gla.ac.uk (C.M.); amy.flaxman@ndm.ox.ac.uk (A.F.); katie.ewer@ndm.ox.ac.uk (K.E.); sarah.gilbert@ndm.ox.ac.uk (S.G.)

**Keywords:** MVA, influenza, vaccination, elderly, nucleoprotein

## Abstract

Background: Pre-existing T cell responses to influenza have been correlated with improved clinical outcomes in natural history and human challenge studies. We aimed to determine the efficacy, safety and immunogenicity of a T-cell directed vaccine in older people. Methods: This was a multicentre, participant- and safety assessor-blinded, randomised, placebo-controlled trial of the co-administration of Modified Vaccinia Ankara encoding nucleoprotein and matrix protein 1 (MVA-NP+M1) and annual influenza vaccine in participants ≥ 65. The primary outcome was the number of days with moderate or severe influenza-like symptoms (ILS) during the influenza season. Results: 846 of a planned 2030 participants were recruited in the UK prior to, and throughout, the 2017/18 flu season. There was no evidence of a difference in the reported rates of days of moderate or severe ILS during influenza-like illness episodes (unadjusted OR = 0.95, 95% CI: 0.54–1.69; adjusted OR = 0.91, 95% CI: 0.51–1.65). The trial was stopped after one season due to a change in the recommended annual flu vaccine, for which safety of the new combination had not been established. More participants in the MVA-NP+M1 group had transient moderate or severe pain, redness, and systemic responses in the first seven days. Conclusion: The MVA-NP+M1 vaccine is well tolerated in those aged 65 years and over. Larger trials would be needed to determine potential efficacy.

## 1. Introduction

Seasonal influenza has a significant global impact, as every year there are an estimated 1 billion cases, 3–5 million severe cases and 290,000–650,000 influenza-related respiratory deaths worldwide [1], with an estimated annual economic cost of $87.1 billion in the US alone [2]. Influenza pandemics occur from time to time in addition to annual seasonal influenza, with a considerable health and economic burden [3]. Vaccination, the most cost-effective strategy available to combat influenza, works by inducing strain-specific antibodies against the highly influenza polymorphic surface proteins, haemagglutinin and neuraminidase. As the circulating virus strains change, vaccines need to be reformulated annually to match new strains arising through genetic drift in the surface proteins.

In situations in which individuals are exposed to a new influenza virus strain against which they lack protective neutralising antibodies, cross-reactive T-cells against conserved internal antigens of influenza have been shown to be associated with less viral shedding, reduced duration of symptoms and less severe symptoms [4,5]. A vaccine against influenza that induced protective T-cell responses against conserved internal antigens could therefore provide longer lasting immunity against not only human seasonal influenza, but also other subtypes currently found in avian or other species, which have the potential to cause a new pandemic [6]. The internal proteins of the influenza virus, such as nucleoprotein (NP) and Matrix protein (M1), are more conserved compared to the external surface glycoproteins. H3N2 NP is 92% and 91% identical between H1N1 and H5N1 strains, respectively. Likewise, H3N2 M1 is 95% and 93% identical between H1N1 strains and H5N1 strains, respectively [7]. A T cell response to NP and M1 can be detected in more than 70% of individuals [8], and recent studies have also shown that these T-cells specific responses were associated with limiting influenza viral shedding, reduced duration of symptoms and minimising severity of symptomatic illness [4,5].

The recombinant viral vector Modified Vaccinia Ankara (MVA) has been used to generate strong T-cell responses to a wide range of antigens, including antigens from *Plasmodium falciparum* [9], *Mycobacterium*
*tuberculosis* [10], hepatitis C virus [11], human immunodeficiency virus (HIV) [12,13] and influenza virus [7,14]. MVA-NP+M1 is a recombinant, replication-deficient MVA vector expressing the influenza antigens NP and M1 as a fusion protein [7]. It has been studied in a large number of small Phase 1 and Phase 2a trials, when manufactured in either chicken embryo fibroblast cells or in the duck immortalized AGE1.CR.pIX cell line [7,14,15,16,17,18,19]. Vaccination with MVA-NP+M1 results in a rapid increase in influenza-specific cross-reactive interferon gamma (IFN-γ)-secreting effector T-cells across age groups, which are maintained at levels above baseline responses over the course of a year [7]. In the older age groups, MVA-NP+M1 can boost pre-existing levels of influenza-specific T-cells and maintain them for up to at least 6 months post-vaccination [17], and co-administration of MVA-NP+M1 with inactivated influenza vaccine to healthy adults did not blunt the increase in T cell responses [15].

To test the ability of the MVA-NP+M1 to alter influenza disease course post-infection, we conducted a pragmatic field trial in elderly adults receiving quadrivalent influenza vaccine (QIV) in the Thames Valley region of the UK to assess the safety and efficacy of the vaccine to reduce the number of days of moderate to severe symptoms during an influenza-like illness (ILI) episode.

## 2. Methods

Invictus (A Phase IIb Study to Determine the Safety and Efficacy of Candidate INfluenza Vaccine MVA-NP+M1 in Combination with Licensed InaCTivated Influenza Vaccine in AdUltS aged 65 years and above) was a multicentre, participant- and safety assessor-blinded, randomised, placebo-controlled study to assess the efficacy and safety of co-administration of MVA-NP+M1 and the recommended licensed quadrivalent IIV in participants aged 65 and above.

Participants: Eligible potential participants were identified by their general practitioners (GP) at six primary care General Practices and one University-based Phase 1 Unit (Centre for Clinical Vaccinology and Tropical Medicine) for the immunology sub cohort (Appendix A) and contacted by letter or advertisement. Inclusion and exclusion criteria are listed in Appendix A. The participants in the main phase attended for a single visit on Day 0 to receive both QIV and the study vaccine, and were followed up over the remainder of the influenza season without specific scheduled visits. Fifty participants were selected to participate in an immunology sub-cohort.

On Day 0, eligible participants were randomised in a 1:1 ratio to receive either 1.5 × 10^8^ pfu of MVA or saline placebo using a web-based system that employed a non-deterministic minimisation on practice, age and gender to ensure each arm was balanced. Ethnicity; race; body mass index (BMI); concomitant medications and co-morbidities, including cardiovascular, metabolic, neurological, eye/ear conditions respiratory, renal or genitourinary and musculoskeletal disease and cancers, were recorded. Fluzone^®^ (Sanofi-Pasteur) inactivated QIV was administered in the non-dominant arm, and the experimental vaccine or placebo was administered randomized 1:1 intramuscularly (IM) within 1 cm of the same site. Participants in the immunology sub-cohort also had pre-vaccination safety laboratory and immunogenicity blood samples taken. After both vaccinations were performed, the vaccination area was covered with a sterile dressing and participants were observed for at least 10 min.

Participants were provided with and instructed in the use of web-based electronic or paper diary, and the end date for self-recorded symptoms was either 30 April or that determined by national influenza surveillance, whichever was later. For the first 7 days after vaccination, the local and systemic reactogenicity were solicited. Unsolicited adverse events were recorded for a further 28 days. Participants were asked if they had any ILS- feverishness, cough, sore throat, generally unwell, headache, muscle ache, shortness of breath and temperature) daily during Weeks 2, 3 and 4, and thereafter weekly if they had any of these symptoms for the rest of the influenza season. Every 3 to 4 weeks, participants were contacted by telephone to enquire about occurrence of any ILS or serious adverse events (SAEs). ILS not recorded in the diary were collected during the telephone call. Any hospitalisations, accident or emergency room visits and GP consultations were also recorded during and at the conclusion of the study. If ILS occurred, participants were asked to record both the nature and severity of their symptoms daily and any medications taken to treat these symptoms.

Study definitions and endpoints: ILI was defined as having a fever or feeling feverish AND a cough and/or sore throat. The start of an ILI was any ILI symptom that occurred during a period in which at least one day with a defined ILI occurred. Seven days without a symptom defined a new ILI episode. In addition, a participant perceived definition of an ILI was defined as being any episode in which any ILS was reported, regardless of whether the ILI protocol definition was met (as the original powering calculation using data from FluWatch [20] used this data set rather than those symptoms occurring during a protocol-defined ILI). A binary variable was generated for each day of follow-up to indicate whether the participant had a moderate or severe influenza-like symptoms, no moderate or severe influenza-like symptom or missing response (missing).

Immunology analysis: Participants in the immunology cohort returned on Day 7, 21 and 182 for safety assessments and immunology blood sample collection. T cell responses were evaluated using ex vivo interferon gamma (IFN-γ Enzyme-linked Spot (ELISPOT) assays and flow cytometry with intracellular cytokine staining assays, as previously described [7]. The number of responders by ELISPOT were assessed for the NP and M1 pools combined using pre-defined criteria, and reported as spot forming units (SFU) per 10*6 peripheral blood mononuclear cells. Serum samples were analysed at Viroclinics (Viroclinics, Rotterdam, Netherlands) for hemagglutinin inhibiting (HAI) antibodies to the four strains of influenza A and B present in the administered QIV.

Statistical analysis: Data collected from FluWatch [4,20] indicated that the average total number of days spent with moderate or severe ILS in vaccinated people in the UK of this age group was 3.5 days per ‘season’ of circulation. Assuming a typical follow-up period for a season of approximately 120 days across participants, this average corresponds to 2.92% of days. A reduction of 1 day per season with moderate or severe ILS corresponded to a 20% relative drop in the proportion of days with moderate or severe ILS to 2.34%. A total of 2030 participants (1015 per group) would therefore provide 85% power to detect the assumed relative 20% drop, at the 5% significance level. This sample size accounted for a 25% attrition rate and a further 15% increase to account for clustering of participants within households (based on FluWatch analyses).

For the secondary outcome that assessed the incidence of ILI between the two groups, data collected from FluWatch estimated that 12.25% of vaccinated individuals over 65 years of age experienced an ILI, most likely attributable to influenza virus infection rather than illness due to other respiratory viruses that was not influenza during the winter season (December to March) [20]. The proposed sample size had 90% power to observe a reduction in the percentage of individuals experiencing ILI from 28.5 to 22.25%, assuming that approximately 43% of ILI is due to influenza. No virologic endpoints were planned for the first year of the study. The study was terminated early by the sponsor after the first 2017/2018 influenza season due a new recommendation by the UK Government, which directed the National Health Service to use an MF59-adjuvanted trivalent influenza vaccine (Fluad^®^) in the 65 year old and above population in the vaccination programme for the planned 2018/2019 influenza season. The rationale for early termination was the absence of any clinical interaction data with MVA-NP+M1 and the trivalent adjuvanted vaccine to allow safe co-administration.

The following were derived from the patient diaries: the number of days with moderate or severe ILS during ILI (the primary endpoint), incidence of ILI and duration of ILI and severity of ILI symptoms. The comparisons between the two vaccine groups in the study were presented using effect measures (ratio or difference in response rate, difference in means) with 95% confidence intervals at a 5% two-sided significance level. The number of days with moderate or severe symptoms were analysed using generalised mixed effect models with logit link, and a random effect included to account for the clustering of data over days within participant. A sensitivity analysis was performed with adjustment for co-morbidities at baseline. The incidence of ILI and occurrence of general practitioner (GP) consultations due to respiratory illness were assessed using a Poisson regression model. Normality assumptions were not met for the duration of ILI; therefore this was log_10_ transformed and analysed using a fixed effect model (with random effect for participant included). Hospitalisations and deaths due to adverse events, and safety post-vaccination observations were compared using a Fisher’s exact test. Solicited adverse events in the 7 days post-vaccination were analysed descriptively. The frequency of influenza infection using historical data on the proportion of ILIs that is caused by influenza virus infection was estimated from the incidence of ILI multiplied by the estimate of the proportion of ILIs that is caused by influenza virus infection derived from historical data.

Ethics and oversight: The study was granted approval by the South Central Berkshire Research Ethics Committee following review of the protocol, the participant information and consent forms and other required documents. All participants signed the written informed consent. The study was registered on clinicaltrials.gov (NCT03300362) on 26 September 2017. The study was reviewed and approved by the Centre for Clinical Vaccinology and Tropical Medicine and the Oxford University Hospitals National Health Service (NHS) Foundation Trust Genetic Modification Safety Committees, as well as the Medicines and Healthcare products Regulatory Agency. The study was conducted in accordance with the Declaration of Helsinki, the International Council on Harmonisation of Technical Requirements for Registration of Pharmaceuticals for Human Use, and guidelines on Good Clinical Practice (GCP) and other applicable regulatory requirements were followed.

A Data Monitoring and Ethics Committee was appointed to periodically review and evaluate the accumulated study data (for participant safety, study conduct, progress and efficacy) and make recommendations concerning the continuation, modification or termination of the study.

## 3. Results

Enrolment: In the first season, 862 elderly participants were enrolled (Figure 1 Consort Diagram) at six primary care General Practices and one University-based Phase 1 unit (immunology site) throughout the Thames Valley area, UK. The two experimental arms that followed the administration of QIV to all participants were well matched for age, site, gender, other demographics, and co-morbidities (Table 1, Appendix A). Of the 862 participants (ITT), 860 were vaccinated (defined as the Safety Population, Appendix A). The influenza season was determined to cover the period 8 December 2017, until 6 April 2018, and only symptoms recorded during that time were included in the analysis. During the 2017/2018 UK season, the overall incidence of influenza was lower than in recent years, and the percentage of isolates determined to be Influenza A was 42%, although over 20% were not classified as either A or B (https://assets.publishing.service.gov.uk/government/uploads/system/uploads/attachment_data/file/740606/Surveillance_of_influenza_and_other_respiratory_viruses_in_the_UK_2017_to_2018.pdf (accessed on 2 February 2021)).

Safety: MVA-NP+M1 was well tolerated in this population of 860 participants aged 65 years and above. Immediate post-vaccination symptoms following MVA-NP+M1 in combination with QIV were all mild in severity, and anaphylaxis, bronchospasm and angioedema were absent. Solicited local and systemic reactogenicity responses over the 7 days post-vaccination were as would be expected and were generally mild in severity (Figure 2). There were more participants with moderate or severe pain, redness and systemic responses in the MVA-NP+M1 group (*p* < 0.01), but these generally resolved to the same level as the placebo within 2 to 3 days (Appendix A). A total of 120 (14.0%) participants reported a total of 208 unsolicited adverse events (108 with MVA-NP+M1 and 100 with placebo, NS) in the 28 days post-vaccination (Appendix A).

All preferred terms occurred with very low incidence (<2.0% participants within a study vaccine group and overall). There was no statistically significant difference between the two study vaccine groups in terms of number of participants with at least one adverse event on Chi-squared test (*p* = 0.8655) and no apparent difference between the two study vaccine groups for any adverse event term. A total of 10.9% participants had at least one severe (Grade 3) adverse event. All severe (Grade 3) preferred terms occurred with very low incidence (<2.0% participants in the MVA-NP+M1 group and <1.5% overall). There were 28 SAEs in 26 participants (15 participants in the MVA-NP+M1 group and 11 participants in the placebo group), and none was assessed to be study vaccine-related. The SAE had a fatal outcome in four participants (two in the MVA-NP+M1 group and two in the placebo group). All four occurred >60 days post-vaccination. In the immunology sub-cohort, there were some fluctuations in mean laboratory safety values over time, and isolated shifts in individual participants, but no apparent differences between the two study vaccine groups.

Outcomes: There was no difference in the reported rates of days of moderate or severe influenza-related symptoms during ILI episodes (Table 2, unadjusted OR = 0.95, 95% CI: 0.54–1.69; adjusted OR = 0.91, 95% CI: 0.51–1.65) (also Appendix A).

The number of cases and participants with self-reported ILI was 83 in 72 participants in the in the MVA group and 94 episodes in 79 participants in the placebo group (Table 3).

These results were essentially unchanged when using the patient perceived ILI definition. There was no difference in the duration of symptoms (Table 3 and Appendix A), rates of GP consultations, hospitalizations or deaths. The severity of individual symptoms did not vary between groups, with trends favouring MVA.

Immunogenicity: The median response to QIV plus placebo in the overall ELISpot response compared to baseline using overlapping peptides to NP and M1 showed no fold change (Figure 3, mean 339 +/− 299 (SD) spot forming units (SFU)/10^6^ PBMCs at day 0 (n = 23), 309 +/− 244 at day 21 (n = 23), 426 +/− 417 at day 182 (n = 24)), whereas the increase was nearly 3-fold in the MVA-NP+M1 arm at day 21 (*p* = 0.0002) (396 +/− 809 at day 0 (n = 24), 923 +/− 737 at day 21 (n = 26), 554 +/− 574 at day 182 (n = 26)).

This data was confirmed in the ICS assays (conducted blinded on a smaller number of participants chosen by elevated ELISpot results, in which significant increases were observed in polyfunctional CD4+ T cells following M1 peptide, and both CD4+ and CD8+ to NP pool stimulation, between baseline and day 21 in the MVA-NP+M1 vaccinated group (Appendix A).

A small previous study in which co-administration of trivalent inactivate influenza vaccine (TIV) with MVA-NP+M1 found increased hemagglutinin antibody responses to some of the TIV components [15] but we did not observe this increase in this elderly cohort (Figure 4, Table 4). There was no statistical difference in the fold-increases in the titers between the Placebo and MVA recipients (all had received IVI) at either day 21 or Day 182.

## 4. Discussion

INVICTUS was a Phase IIb study of the MVA-NP+M1 vaccine, an MVA vector expressing NP and M1 from the influenza A virus (H3N2; A/Panama/2007/99) as a single fusion protein. The purpose of the overall clinical programme is to develop a more efficacious vaccine capable of providing protection against a broad spectrum of influenza A virus strains, with emphasis on producing better CD4^+^ and CD8^+^ T cell responses in those at higher risk of severe influenza disease. Older people are at particularly higher risk, with vaccination with the currently available licensed vaccine preventing only 30 to 40% of laboratory-confirmed influenza [21]. The main objective of the study was to investigate the efficacy of MVA-NP+M1 compared to placebo, in combination with the annual licensed IIV in an adult population aged 65 years and above. Other objectives included assessment of safety and reactogenicity and vaccine cellular and humoral immunogenicity.

A total of 431 participants with MVA-NP+M1 and 429 with placebo were vaccinated in the first season of the study. The study was terminated early after this first influenza season due to the changes recommended by the UK Government which directed the NHS to use an MF59-adjuvanted trivalent influenza vaccine (FLUAD^®^; Sequirus UK Ltd.) in the 65 year old and over population in the second season. No safety data on co-administration of this vaccine with MVA-NP+M1 was available, with insufficient time to study the combination prior to the second season. A blinded analysis performed by the DMC at the end of the first season also indicated the study was significantly likely underpowered, potentially exacerbated during a majority influenza B circulating year, even if the full cohort could be enrolled in the second season. The MVA-NP+M1 (Panama H3N2) vaccine has approximately only 20% conservation at the amino acid level to influenza B viruses, and the moderate 2017/2018 UK influenza season made up 38% of influenza B isolates compared to 42% for influenza A (with equivalent numbers of hospitalizations and visits). In addition, the UK region of the study had a lower rate of documented influenza than the overall country average.

There was no apparent difference between the MVA-NP+M1 and placebo groups in terms of the primary endpoint (number of days with moderate or severe influenza-like symptoms during an ILI episode) or other efficacy endpoints (incidence and duration of ILI, severity of symptoms or GP consultations, hospitalisations or deaths due to respiratory illness); however, due to the powering issue, no efficacy conclusions, positive or negative, can be drawn. Thus, the increase in local side effects may have no associated benefit. There was a marked vaccine-specific T cell response in this elderly population, as measured by IFN-γ ELISPOT, detected 21 days post-vaccination that was directed against both antigens in the vector insert.

A small previous study showed enhancement of HAI antibody responses to the H3N2 and H1N1 component of the TIV vaccine, and this fact, along with safety of co-administration, was a basis for the design [15]. In this, 65 years and over age group vaccination with MVA-NP+M1 did not increase influenza-specific antibody responses to the influenza strains in the licensed seasonal vaccine. Whether this is due to the overall poor responses generated by this QIV in the elderly or a lack of adjuvant effect is not clear. However, the finding has led further development to administer the MVA-NP+M1 at a separate site, and potentially a different time, than the recommended inactivated vaccine.

MVA-NP+M1 was well tolerated, with a local and systemic reactogenicity profile predominantly of mild and moderate self-limited adverse events which represented typical post-vaccination reactions and were comparable to those seen in the previous smaller studies [7,14,15,16,17,18,19]. The incidence of SAEs was low for a population age 65 or older (3.5% of participants in the MVA-NP+M1 group), was comparable with placebo and no SAE was assessed to be study vaccine-related.

The study was designed without virologic endpoints to establish efficacy on clinical outcomes and to evaluate patient experience of illness. Of note, incidence and duration of moderate or severe symptoms is not an accepted regulatory endpoint for licensure of a novel influenza vaccine. Large community efficacy studies in which the standard primary endpoint of laboratory-confirmed influenza has been used have rarely observed a statistically significant reduction in ILI, although some nursing home based trials have [22,23,24]. In addition, a recent study found no change in ILI despite an efficacious vaccine, and has brought into question again whether or not there may be “ILI replacement or competition” [25,26]. The symptoms of non-influenza and influenza viral infections are difficult to distinguish in the elderly, and studies have questioned whether rhinovirus and influenza may compete for this illness niche [27,28]. Given the lack of definitive data in this area, it remains imperative to use a virologic endpoint, both to better understand the effect on different circulating strains and document biologic activity.

The need to suspend the study after one season led the sponsor to re-design the follow-on study to use all age groups, not co-administer the vaccine, and to collect virologic endpoints to better understand the effect of the vaccine on various circulating strains. That trial, which planned to enrol 6000 subjects in Australia over two seasons, was started in the following 2019 season.

In conclusion, in this study, MVA-NP+M1 was co-administered with standard QIV to a large group of older people, and induced expected increases in the T cell response. In this underpowered study, there was no benefit seen in this trial to inducing higher levels of HAI antibodies with MVA-NP+M1, and we were unable to show any association of outcomes with gamma interferon T cell responses. A larger study with virologic-confirmed endpoints would be needed to definitively rule out this hypothesis.

## Figures and Tables

**Figure 1 vaccines-09-00851-f001:**
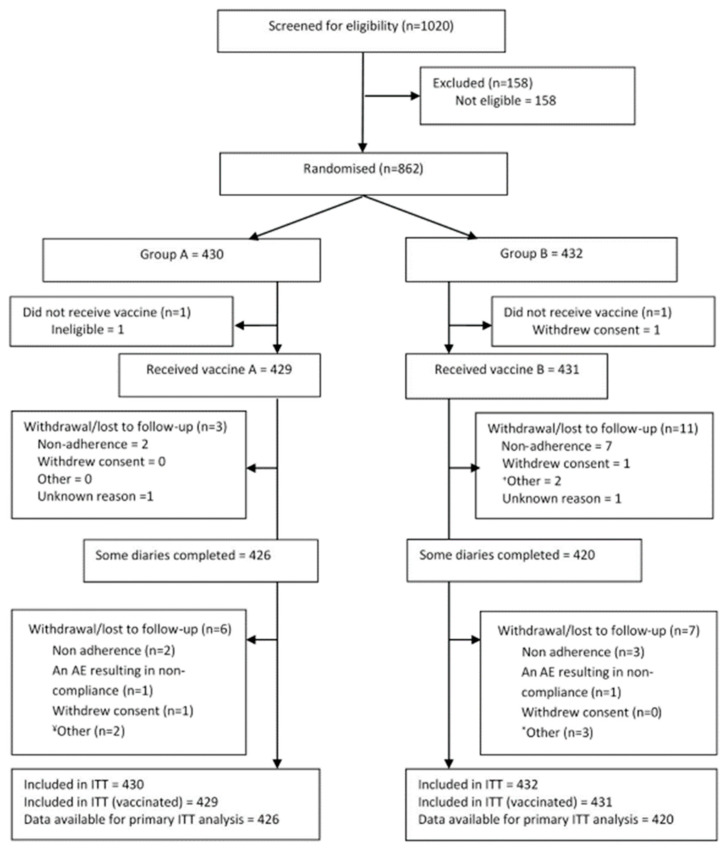
CONSORT Diagram showing patient screening and flow. ITT- intention to treat.

**Figure 2 vaccines-09-00851-f002:**
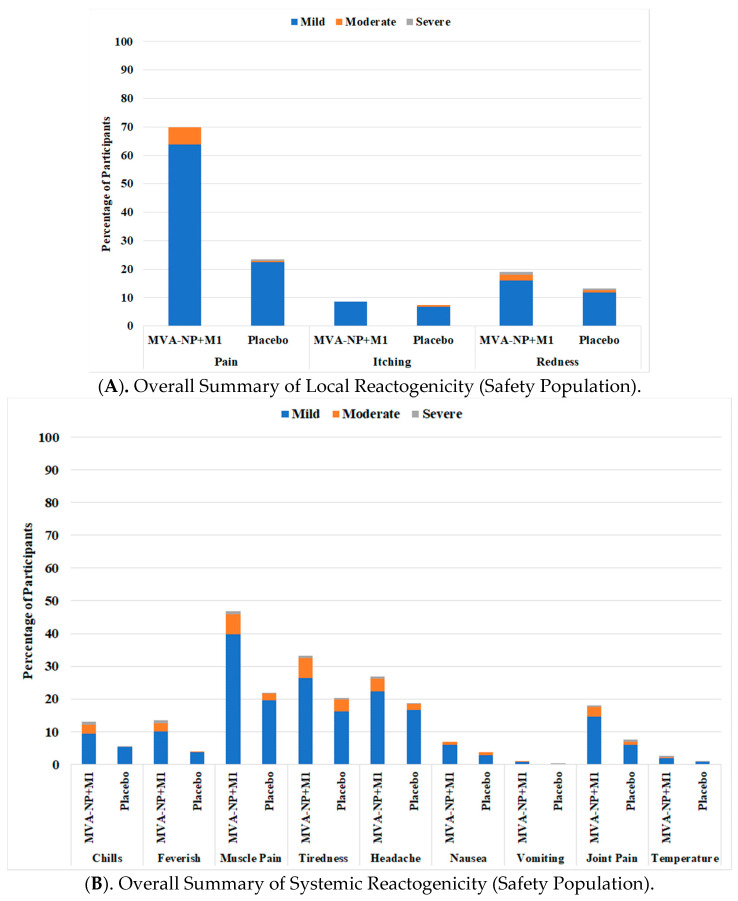
Local (**A**) and Systemic (**B**) reactogenicity in the 7 days following vaccination as recorded on patient diary cards.

**Figure 3 vaccines-09-00851-f003:**
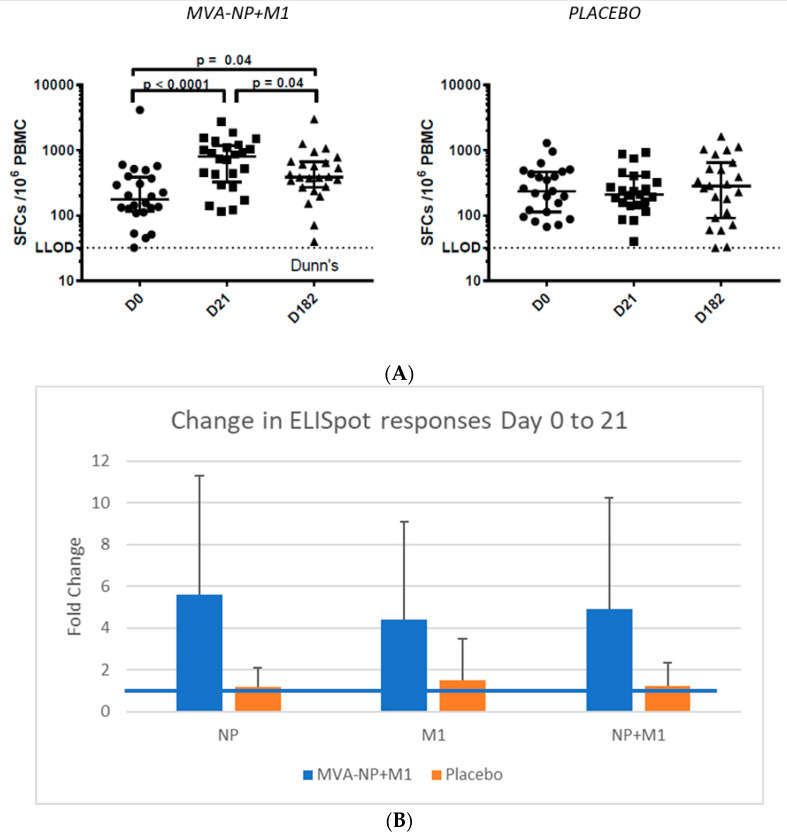
(**A**). Gamma interferon ELISpot responses to NP and M1 pools A. Data over the study shown as median plus 25 and 75% intervals at days 0, 21 and 182. (**B**). Fold change (+SD) in responses at day 0 and 21 to the NP or M1 peptide pools. Increases were significant comparing MVA-NP+M1 to placebo for each antigen (*p* < 0.01). The blue line represents no change (fold increase of 1).

**Figure 4 vaccines-09-00851-f004:**
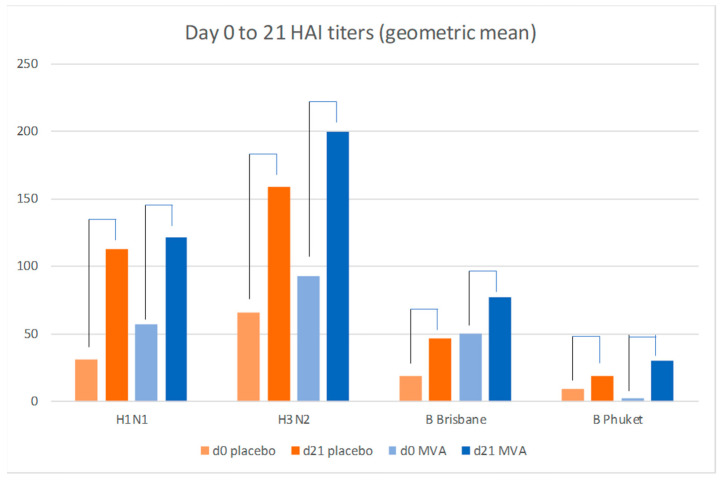
Influenza-specific hemagglutination geometric mean titers to the 2017 vaccine components (H1N1, H3N2, B Brisbane and B Pukhet) in the MVA (*n* = 26) and placebo recipients (*n* = 23) (SD and *p* values shown in Table 4); all participants received inactivated influenza vaccine.

**Table 1 vaccines-09-00851-t001:** Summary of Demographics (ITT Population).

Variable	Statistic	MVA-NP+M1 (*n* = 432)	Placebo (*n* = 430)	Total (*n* = 862)
Age	Mean (SD)	72.5 (5.1)	72.4 (4.9)	72.4 (5.0)
Gender *n* (%)	Male	240 (55.6%)	237 (55.1%)	477 (55.3%)
	Female	192 (44.4%)	193 (44.9%)	385 (44.7%)
Race *n* (%)	Non-white	5 (1.2%)	2 (0.5%)	7 (0.8%)
	White	426 (98.6%)	428 (99.5%)	854 (99.1%)
Height (cm)	Mean (SD)	169.8 (9.6)	170.1 (9.6)	170.0 (9.6)
Weight (Kg)	Mean (SD)	78.0 (16.3)	78.2 (16.4)	78.1 (16.4)

**Table 2 vaccines-09-00851-t002:** Primary Outcome (ITT). Unadjusted and Adjusted Linear Mixed Effect Model for Presence of Moderate or Severe Symptoms.

-	Odds Ratio	*p*-Value	95% Lower CI	95% Upper CI
Unadjusted (*n* = 846)	-	-	-	-
Treatment group (MVA-NP+M1 versus Placebo)	0.954	0.872	0.537	1.694
Adjusted *(n* = 845)	-	-	-	-
Treatment group (MVA-NP+M1 versus Placebo)	0.914	0.766	0.508	1.647

Abbreviations: CI = confidence interval. [a] Adjusted for age, gender, co-morbidities and with participant identifier fitted as a random effect (n = 1, missing age).

**Table 3 vaccines-09-00851-t003:** Summary of Secondary Efficacy Endpoints (ITT Population).

	MVA-NP+M1		Placebo	
	N	Summary	N	Summary
Number (%) participants with an ILI	420	72 (17.1%)	426	79 (18.5%)
Number (%) of ILI episodes per participant	-	-	-	-
0	-	348 (82.9%)		347 (81.5%)
1	-	62 (14.8%)		67 (15.7%)
2	-	9 (2.1%)		9 (2.1%)
3	-	1 (0.2%)		3 (0.7%)
≥4	-	0 (0.0%)		0 (0.0%)
Mean (SD) days of moderate/severe influenza-like symptoms	-	-	-	-
Cough	420	1.2 (3.9%)	426	1.2 (3.3%)
Sore throat	420	0.4 (1.9%)	426	0.5 (1.5%)
Generally unwell	420	0.9 (2.5%)	426	1.0 (2.6%)
Feverishness	420	0.2 (1.1%)	426	0.2 (0.9%)
Headache	420	0.4 (2.0%)	426	0.4 (1.5%)
Muscle Ache	420	0.4 (1.7%)	426	0.3 (1.5%)
Shortness of Breath	420	0.3 (1.8%)	426	0.4 (2.1%)
Temperature	420	0.0 (0.1%)	426	0.0 (0.3%)
Median [Range] duration of symptoms per protocol defined ILI episode	83	7.0 [1.0 to 58.0]	94	5.0 [1.0 to 28.0]
Number of GP consultations for respiratory illness		66		42
Number (%) participants with at least one General Practitioner consultation from respiratory illness	431	44 (10.2%)	429	35 (8.2%)
Number (%) participants with at least one hospitalisation and/or death due to respiratory illness	431	3 (0.7%)	429	3 (0.7%)

**Table 4 vaccines-09-00851-t004:** Increase in geometric mean titer between Day 0 and Day 21 in the immunogenicity sub-study for the two experimental groups.

Day 21	A/Michigan/45/20152	A/HongKong/4801/20143	B/Brisbane/60/20084	B/Phuket/3073/20135
MVA-NP+M1 (*n* = 26)	5.48 (13.32)	3.91 (5.72)	2.75 (6.16)	1.92 (1.66)
Placebo (*n* = 23)	12.23 (28.48)	14.92 (53.00)	13.05 (45.75)	4.48 (7.21)
*p* value	0.29	0.31	0.27	0.09
**-**	-	-	-	-
Day 182	-	-	-	-
MVA-NP+M1 (*n* = 26)	3.83 (12.61)	2.77 (3.76)	1.56 (1.42)	1.36 (0.97)
Placebo (*n* = 23)	4.37 (9.79)	4.68 (13.09)	4.4 (11.42)	2.17 (23.56)
*p* value	0.87	0.49	0.22	0.3

## Data Availability

The data for this study may be requested from Chris Butler at the contact address listed. Please send a summary of the analysis planned and the data required for consideration.

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
