# Peer review of "Efficacy and Safety of a Modified Vaccinia Ankara-NP+M1 Vaccine Combined with QIV in People Aged 65 and Older: A Randomised Controlled Clinical Trial (INVICTUS)"

_vaccines, 2021, doi:10.3390/vaccines9080851_

Round 1

Reviewer 1 Report

In this manuscript, the authors describe a phase IIb clinical trial that combines the seasonal QIV with that of a T cell stimulating vaccine (MVA+NP+M1) in elderly adults. The rationale and background are evident and the study is nicely laid out. The overall results are interesting, but not surprising. Overall, the manuscript is written and organized well. The only major adjustments needed are most of the figures are not publication quality and several additional analyses of existing or potentially existing data would be helpful for interpretation. Specific comments are listed below. 

Major Comments:

  1. The authors include a statistics section in the methods, but there is a general lack of statistics throughout the manuscript within individual figures, notably Figures 3 and 4.
  2. Additionally, it’s not clear what the number of participants were in Figures 3 and 4. There is also a lack of error bars in Figure 4.
  3. It would be more revealing to show a fold-change in serum HAI titers, similar to Figure 3B.
  4. Were HAI titers at d182 tested? Similarly, were total titers against the vaccine tested?

Minor Comments:

  1. Reference 1 is out of date and I’m not sure I follow the 1 billion illnesses number cited. Surely the authors aren’t suggesting 1 billion people have an influenza illness each year.
  2. Paragraph 3 of the introduction – Plasmodium should be capitalized and include something along the lines of Plasmodium parasites. Formal name should be included for tuberculosis and virus should be added to hepatitis C and influenza.
  3. Figure 3A – it would be clearer to have the placebo and experiment group results right next to each other.
  4. Figure 3B – the authors should draw a line across the graph at 1, as that would indicate no change. This is particularly important as it’s clear the placebo group would almost fall exactly on that line.
  5. Figures are generally low quality, mostly low resolution and the use of gray text.

Author Response

Major Comments:

  1. The authors include a statistics section in the methods, but there is a general lack of statistics throughout the manuscript within individual figures, notably Figures 3 and 4.

Statistical results are now included for both figures in the text or accompanying Table.

  1. Additionally, it’s not clear what the number of participants were in Figures 3 and 4. There is also a lack of error bars in Figure 4.

We have not added the error bars to Figure 4, as it would clutter the figure, but the P value and numbers are shown in adjacent Table 4. The numbers in Figure 3 are now stated in the text.

  1. It would be more revealing to show a fold-change in serum HAI titers, similar to Figure 3B.

The change in HAI titers

These are now shown in Table 4.

  1. Were HAI titers at d182 tested? Similarly, were total titers against the vaccine tested?

Day 182 is now also listed in Table 4, whereas Figure 4 gives an idea of the absolute values. Standard deviations are now also included, along with p values.

Titers against the vaccine inserts were not tested, as there are no data that titers to the internal proteins of influenza play a role in protection, and since no protection was clearly seen, these were not seen to be of importance. Due to this fact Vaccitech has also not developed a qualified assay for titers to  either M1 or NP.

Minor Comments:

  1. Reference 1 is out of date and I’m not sure I follow the 1 billion illnesses number cited. Surely the authors aren’t suggesting 1 billion people have an influenza illness each year.

A new and up to date reference is added. It is not true that there are 1 billion illnesses, but it is true that influenza infects up to 20% of the world population (or greater than 1 billion CASES) per year. This has been corrected and a more recent URL reference added.

  1. Paragraph 3 of the introduction – Plasmodium should be capitalized and include something along the lines of Plasmodium parasites. Formal name should be included for tuberculosis and virus should be added to hepatitis C and influenza.

These have been done.

  1. Figure 3A – it would be clearer to have the placebo and experiment group results right next to each other.

We believe the figure is clear as presented, although appreciate the suggestion.

  1. Figure 3B – the authors should draw a line across the graph at 1, as that would indicate no change. This is particularly important as it’s clear the placebo group would almost fall exactly on that line.

This line has been added.

  1. Figures are generally low quality, mostly low resolution and the use of gray text.

We have uploaded new figures and changed the gray text to black.

Reviewer 2 Report

In this short article, Butler et al. concluded that MVA-NP+M1 can be safely co-administerd with licensed quadrivalent influenza vaccine. The data suggest that the experimental vaccine can elicit stronger T cell responses. However, no increase in antibody titers were observed. Lastly, the experimental vaccination did not exhibit improved efficacy. My major concern has to do with presentation and statistical analyses and they are outlined below:

  1. Figure 2A. No statistical significance is indicated but the difference between the vaccine group and placebo group for pain seems significant (70% vs 23%: more than 3-fold difference). How were the results analyzed? Similarly, it is surprising that no statistical significance was achieved for Figure 2B. There is a two-fold increase in muscle pain for the vaccinated group.
  2. Is there a statistical significance in Figure 3B? The error bar is so wide that it is unlikely. If so, the authors conclusion is not supported by the results.
  3. The supplemental Figure 2 is more convincing that the main figure. I would suggest putting the S. Fig 2 in the main figure.
  4. In many figures, the control group is labelled as “Placebo” but they are getting QIV. This is misleading.
  5. Figure 4. The figure legends need more details. What is the sample size? What statistical analyses were used? Where are the error bars?
  6. Figure 4. Was statistics done on this figure? There are multiple >2 fold differences in this figure. Further, there appears to be a difference in HAI titer between d21 MVA vs d21 placebo for H3N2 and B birsbane.
  7. Figure 4. Why are the baseline HAI titer higher in MVA groups, with exception of HAI against B Phuket (d0 MVA vs. d0 placebo)?
  8. What is the rationale for co-administering an experimental vaccine with currently used TIV? TIV itself can provide good protection when antigens are well-matched. This leaves very small window for the experimental vaccine to show its efficacy. It would make more sense to compare MVA-NP+M1 vs. TIV vs. placebo. Please provide justification for combining MVA-NP+M1 with TIV in introduction.

Minor comments:

  1. The submitted manuscript does not contain continuous line numbers.
  2. Several typos were found.
  3. Abstract: Define “ILS” in the abstract.
  4. 3 Immunology analysis: correct “IFN-???ELISPOT”
  5. The resolution of figure 1 is poor.
  6. Supplementary Figure 1&2. Figure legends are missing. What are the red and blue lines in s. fig 1? The font is too small and not readeable.

Author Response

In this short article, Butler et al. concluded that MVA-NP+M1 can be safely co-administered with licensed quadrivalent influenza vaccine. The data suggest that the experimental vaccine can elicit stronger T cell responses. However, no increase in antibody titers were observed. Lastly, the experimental vaccination did not exhibit improved efficacy. My major concern has to do with presentation and statistical analyses, and they are outlined below:

  1. Figure 2A. No statistical significance is indicated but the difference between the vaccine group and placebo group for pain seems significant (70% vs 23%: more than 3-fold difference). How were the results analyzed? Similarly, it is surprising that no statistical significance was achieved for Figure 2B. There is a two-fold increase in muscle pain for the vaccinated group.

The reviewer is correct that there is a statistically significant difference, however safety data is not presented as a statistical consideration but as results only.

  1. Is there a statistical significance in Figure 3B? The error bar is so wide that it is unlikely. If so, the authors conclusion is not supported by the results.

The p value for this figure has been added. (see comments form Reviewers 1 and 2 as well)

  1. The supplemental Figure 2 is more convincing that the main figure. I would suggest putting the S. Fig 2 in the main figure.

This is confusing as the two Figures pointed out represent different data.

  1. In many figures, the control group is labelled as “Placebo” but they are getting QIV. This is misleading.

All subjects received QIV and then were randomized to either placebo or MVA-NP+M1. This is clearly stated in the methods, so the experimental group is correctly labelled placebo.

  1. Figure 4. The figure legends need more details. What is the sample size? What statistical analyses were used? Where are the error bars?

Figure 4 has been supplemented with Table 4, and please see the new legend.

  1. Figure 4. Was statistics done on this figure? There are multiple >2 fold differences in this figure. Further, there appears to be a difference in HAI titer between d21 MVA vs d21 placebo for H3N2 and B birsbane.

We have added the statistics, which are based on the increase from day 0 to day 21, and there is no significant difference (see Table 4)

  1. Figure 4. Why are the baseline HAI titer higher in MVA groups, with exception of HAI against B Phuket (d0 MVA vs. d0 placebo)?

Please note that this sample is based on a mere 50 subjects enrolled into the immunology subset (25 receiving placebo and 25 receiving MVA-NP+M1), and thus the baseline variability when analyzing four different antigens is not unexpected.

  1. What is the rationale for co-administering an experimental vaccine with currently used TIV? TIV itself can provide good protection when antigens are well-matched. This leaves very small window for the experimental vaccine to show its efficacy. It would make more sense to compare MVA-NP+M1 vs. TIV vs. placebo. Please provide justification for combining MVA-NP+M1 with TIV in introduction.

There are data (references) that individuals with higher T cell responses are better protected entering into a flu season, even when they have received influenza vaccine (IVI). In addition, in a small initial study, there appeared to be an adjuvant activity that showed boosting of HAI titers in younger participants receiving the IVI and the MVA in the same location, which is the method used in this study.  

Minor comments:

  1. The submitted manuscript does not contain continuous line numbers.

This was not requested by mdpi, but we are happy to add, if desired.

  1. Several typos were found.

We have rechecked and used UK spelling.

  1. Abstract: Define “ILS” in the abstract.

Done

  1. 3 Immunology analysis: correct “IFN-???ELISPOT”

Issue with changing computers, we have spelled out gamma so that this will be clear.

  1. The resolution of figure 1 is poor.

We have uploaded a higher resolution figure.

  1. Supplementary Figure 1&2. Figure legends are missing. What are the red and blue lines in s. fig 1? The font is too small and not readable.

Figure legends are now found at the top of each Figure.

Reviewer 3 Report

This is an article describing Efficacy and Safety of a Modified Vaccinia Ankara-NP+M1 Vaccine Combined with QIV in people aged 65 and older. After a well-focused Introduction, the Materials and Methods section provides proper information on the approaches. Statistical analyses were carried out. In my opinion, the manuscript by Butler et al. is free of major weaknesses.

Remarks are as follows:

1) The title contains the abbreviation INVICTUS. The same for the Methods chapter (Invictus). However, this abbreviation is not explained fully. Is it correct that it stands for "Improved Novel VaccIne CombinaTion InflUenza Study". Please clarify this issue. Some other abbreviations (e.g. ILS, ILI) are not described at their first occurrence. Some (e.g. QIV, IIV, ITT, GP, IFN-@@@ELISPOT/ELISpot, SFC, PBMC, SFU, ICS, TIV, DMC, KR) are not described at all. Please elaborate this issue.
2) Statistical test applied for Figure 3A is not revealed.

Based on the above, I recommend the publication of this manuscript after revising the manuscript.

Author Response

This is an article describing Efficacy and Safety of a Modified Vaccinia Ankara-NP+M1 Vaccine Combined with QIV in people aged 65 and older. After a well-focused Introduction, the Materials and Methods section provides proper information on the approaches. Statistical analyses were carried out. In my opinion, the manuscript by Butler et al. is free of major weaknesses.

Remarks are as follows:

  • The title contains the abbreviation INVICTUS. The same for the Methods chapter (Invictus). However, this abbreviation is not explained fully. Is it correct that it stands for "Improved Novel VaccIne CombinaTion InflUenza Study”? Please clarify this issue. Some other abbreviations (e.g. ILS, ILI) are not described at their first occurrence. Some (e.g. QIV, IIV, ITT, GP, IFN-@@@ELISPOT/ELISpot, SFC, PBMC, SFU, ICS, TIV, DMC, KR) are not described at all. Please elaborate this issue.

We have added the bold versions of the designated characters at the beginning of the Methods section. We have placed the full name before each of the abbreviations, although many are standard.

2) Statistical test applied for Figure 3A is not revealed.

We have added the statistical value to the Figure 3A in the text

Reviewer 4 Report

This is a well-written and discussed manuscript  of a failed trial of the co-administration of Modified Vaccinia Ankara encoding nucleoprotein and matrix protein 1 (MVA-NP+M1) and the recommended annual influenza vaccine in participants aged 65 and above. The reasons for failure are well explained in the text, as are the prospects of a future clinical trial of this well-known but underused vaccine platform to enhance the immune response in humans. This strategy will, nevertheless, most possibly, have little relevance in the future due to the development of the mRNA platforms. Despite this personal speculative transgression, I fully agree with the opportunity to publish the current study.  

 I have no further comments on the current version of the manuscript. And my recommendation is to accept it for publication in its current form. Minor spell corrections are needed. 

Author Response

I have done a spell check of the final version and left it in UK English, which may be one reason to cite typos. 

Round 2

Reviewer 2 Report

Please correct “, ,”.

The authors responded to the comments that “The reviewer is correct that there is a statistically significant difference, however safety data is not presented as a statistical consideration but as results only.” However, the text states that “There was no statistically significant difference between the incidence of pain, itching, and warmth between the study vaccine groups.” Please clarify. If statistical significance does exist, then this vaccine offers no benefit but only increases incidences of side effects, of which some are moderate to severe.

The authors stated that “The blue line represents no change (fold increase of 1).” – fig 3 legend. No line is shown in the figure.

The authors commented in the rebuttal that “All subjects received QIV and then were randomized to either placebo or MVA-NP+M1. This is clearly stated in the methods, so the experimental group is correctly labelled placebo.” However, within Figure 3, the panel A has a group labelled as “QIV+placebo D..” and the panel B has a group labeled as “Placebo”. Once again this is very misleading.  

 Fig 4: With additional info on statistical data, it is clear that MVA vaccination does not provide any benefit. This needs to be acknowledged and discussed instead of simply stating that further clinical trials are needed.

The gamma interferon data does not correlate well with the protection data. Despite MVA showing higher gamma interferon ELIspot responses, no significant difference was found in terms of the primary endpoint or efficacy endpoints. The protection data is more in line with the HAI titer, a well-established correlate of protection, which shows no statistically significant improvement by MVA vaccination. Overall, the data do not support the MVA vaccination strategy. However, the authors stated in the discussion that “however, due to the powering issue, no efficacy conclusions, positive or negative, can be drawn.” The protection data is clear that MVA does not offer any advantage.

Author Response

The authors responded to the comments that “The reviewer is correct that there is a statistically significant difference, however safety data is not presented as a statistical consideration but as results only.” However, the text states that “There was no statistically significant difference between the incidence of pain, itching, and warmth between the study vaccine groups.” Please clarify. If statistical significance does exist, then this vaccine offers no benefit but only increases incidences of side effects, of which some are moderate to severe.

  • This is now clarified by removal of this sentence, which referred to the unsolicited events. The actual increase in local events is noted in the following sentences, as well as in Figure 2, Supplemental Figure 1 and Supplemental Table 5

The authors stated that “The blue line represents no change (fold increase of 1).” – fig 3 legend. No line is shown in the figure.

  • The line shows up in my uploaded copy, but was faint. The figure has been replaced with a figure with a thicker blue line.

The authors commented in the rebuttal that “All subjects received QIV and then were randomized to either placebo or MVA-NP+M1. This is clearly stated in the methods, so the experimental group is correctly labelled placebo.” However, within Figure 3, the panel A has a group labelled as “QIV+placebo D..” and the panel B has a group labeled as “Placebo”. Once again this is very misleading.  

  • Thank you for picking up this misleading figure. We have changed the figure 3A panel to Placebo alone to MVA-NP+M1 to have consistency between the figures

 Fig 4: With additional info on statistical data, it is clear that MVA vaccination does not provide any benefit. This needs to be acknowledged and discussed instead of simply stating that further clinical trials are needed.

  • The statistical data shown a trend for the MVA, and the original power was not reached, and thus there was no clear benefit shown. However, this does not imply that repowering could not have resulted in the ability to see the 10-20% benefit seen under the trend. Thus we are fine to say that we were not able to prove benefit, but we also cannot rule out benefit if the trial had fully enrolled. In addition, the data from van Beek et al (JID) showed that the symptoms of influenza (the primary endpoint in this trial) may be replaced with other viruses even when a virological incidence benefit is shown, and implies that we may have seen a benefit if virological endpoints had been gathered and used as the endpoint. Therefore, the follow on trial used virologic-confirmed influenza as the primary endpoint, with symptoms severity and length as secondary endpoints. To clarify this, however, we added the sentence: Thus, the increase in local side effects may be with no associated benefit.” In the discussion and changed the abstract to read “Larger trials would be needed to determine potential efficacy” from “Larger trials are needed to determine efficacy”.

The gamma interferon data does not correlate well with the protection data. Despite MVA showing higher gamma interferon ELISpot responses, no significant difference was found in terms of the primary endpoint or efficacy endpoints. The protection data is more in line with the HAI titer, a well-established correlate of protection, which shows no statistically significant improvement by MVA vaccination. Overall, the data do not support the MVA vaccination strategy. However, the authors stated in the discussion that “however, due to the powering issue, no efficacy conclusions, positive or negative, can be drawn.” The protection data is clear that MVA does not offer any advantage.

  • Once again, the trial is underpowered, and thus definitive conclusions cannot be made for either efficacy or lack thereof, and this does not allow assessment of any correlate as there is not an arm without QIV. In trials of challenge with viruses with low HAI titer, T cells have been the primary correlate , which was the motivation for doing these trials. However, to make this clear we have concluded with the words: “In this underpowered study there was no benefit seen in this trial to inducing higher levels of HAI antibodies using MVA-NP+M1, and we were unable to show any association of outcomes with gamma interferon T cell responses. A larger study with virologic-confirmed endpoints would be needed to definitively rule out this hypothesis.”